# Epidemiological Trends of Acute Chemical Poisoning among Children over a Recent Three-Year Period in Saudi Arabia

**DOI:** 10.3390/children10020295

**Published:** 2023-02-02

**Authors:** Mohammed Merae Alshahrani, Hussein Ayed Albogami, Ali Ahmad Asiri, Khaled Saad Al haydhah, Ibrahim Mohammed Aldeailej, Mohammed Abdullah Aldehaim, Mahmoud Yousef Lubbad, Lolwah Abdulaziz Alalyan, Abdullah Fayez Alasmari, Ismail Yahya Al salem, Abdulaziz Alqahtani, Ahmed Abdullah Al Awadh

**Affiliations:** 1Department of Clinical Laboratory Sciences, Faculty of Applied Medical Sciences, Najran University, Najran 61441, Saudi Arabia; 2General Directorate of Environmental Health, Ministry of Health, Riyadh 12431, Saudi Arabia; 3Riyadh Regional Laboratory, Ministry of Health, Riyadh 12746, Saudi Arabia; 4Public Health Agency, Ministry of Health, Riyadh 12282, Saudi Arabia; 5Department of Clinical Laboratory Sciences, College of Applied Medical Sciences, King Khalid University, Abha 62529, Saudi Arabia

**Keywords:** acute chemical intoxications, epidemiology, children, unintentional poisoning, intentional poisoning

## Abstract

Acute intoxication from chemicals is a major medical emergency that can result in illness and mortality. This retrospective study aims to evaluate acute chemical poisoning incidents among children in Saudi Arabia from 2019 to 2021. A total of 3009 children were recorded as being chemically intoxicated. The SPSS/PC statistics package was used for the statistical analysis. The acute chemical poisoning incidents that occurred in the age groups were <1 year: 237 (7.8%), 1–5 years: 2301 (76.4%), 6–12 years: 214 (7.1%), and 13–19 years: 257 (8.5%). The mean rate of acute chemical poisoning in the northern region was 40.1%. The most common poisonous agents were organic solvents (20.4%) and disinfection agents (22.7%). Interestingly, there is a significant relationship between the different types of acute chemical poisoning and various factors, including gender, age, the location where acute chemical intoxication occurred, the type of exposure, and whether these were intentional or unintentional events. The data suggest that the northern region of Saudi Arabia has had the highest number of recorded incidents of acute chemical poisoning over the last three years (2019–2021). Individuals between 1–5 years old were the worst hit. Organic solvents and detergents were to blame for the acute unintentional chemical poisonings that took place in homes. Therefore, educating the public about chemical poisoning and reducing children’s exposure to toxic chemicals requires educational programs, which may help to reduce chemical poisoning occurrence.

## 1. Introduction

Intoxication from chemicals is a condition that affects people all around the globe. It is responsible for significant morbidity and death rates throughout communities, making it one of the most prevalent types of unexpected medical conditions that require immediate attention [1,2]. According to a World Health Organization (WHO) report (2008), an estimated 45,000 deaths among children and young people (under the age of 20) are linked to acute poisoning annually [3]. According to the American Association of Poison Control Centers (AAPCC), more than 1.3 million children were exposed to poisonous substances in 2015, with 40% of them being children under the age of three [4]. A total of 346 acute poisoning cases were registered in the emergency department of the Children’s Hospital and the Institute of Child Health, Lahore. Over five years, from September 1998 to August 2003, pharmaceutical products were the leading cause (51%), followed by petroleum products (23%), chemicals (8.4%), household substances (7.6%), and unidentifiable agents in (10%) [5]. Additionally, many chemicals, particularly pesticides, are used with suicidal intent, accounting for 10 to 20% of global suicides [6,7]. By using data from all six regions, it was calculated that the yearly number of pesticide suicides globally is 258,234, with a credible range of 233,997 to 325,907. This accounts for 30% (range: 27–37%) of all suicides worldwide. If higher estimates of suicides in India are included in the model, the total annual number of global suicides rises from 873,000 to 1,181,200, the estimated number of pesticide-related suicides rises from 371,594 (347,357 to 439,267), and the proportion of global suicides due to pesticide ingestion rises from 29% to 37% [8]. The highest incidence of chemical intoxication was seen in those under the age of 30 years, particularly females [9,10]. According to studies, the mortality rate from poisoning varies depending on the cultural and geographical factors of various communities [11]. The mortality rate in developed countries is around 0.5 per 100,000 people; however, it is much higher in low-income and developing countries, reaching approximately 2.0 per 100,000 people (almost four times higher than the rate in high-income and developed countries) [12,13]. In contrast, it has been shown that there were no death cases reported following acute poisoning in 218 referred children to the emergency room [14]. Only two of them suffered esophageal sequelae, and this might be attributable to the fact that 95.7% of patients arriving at the emergency department within 2–6 h of poisoning [14]. In Saudi Arabia, several investigations on acute pediatric poisoning have been undertaken. An analysis of seven years’ worth of case records of children admitted to a single military hospital in Hafr Al Batin City, Saudi Arabia, found 168 unintentional pediatric poisonings out of 9951 pediatric admissions (1.7%). This was most prevalent in children aged 1 to 3 years (63%), and the most common causes of poisoning were pharmaceuticals (*n* = 108, 64.3%) and household items (*n* = 60, 35.7%), such as cleaning products [2]. Another study in Abha City, Saudi Arabia, examined poisoning trends in 114 children aged <12 years. They observed that the peak age for poisonings in children is before the age of four, with medicinal medications being the most prevalent agents of poisoning. They also noted that the majority of poisoning episodes occurred in living rooms and bedrooms [15]. In 2014, acute poisoning in both adults and children in Al Majmaah City, Saudi Arabia, was examined after reviewing 169 records retrospectively. They recommended the “establishment of legislations to prevent over-the-counter sales of hazardous substances in childproof containers” as well as “improving medical record-keeping for better information access” [16]. The presence or absence of religious, geographical, economic, or cultural settings, as well as the accessibility of xenobiotics, may be responsible for the variations in the patterns and rates of chemical intoxication [17,18]. Due to the challenges of diagnosis and the vast number of xenobiotics that may induce toxicity, the therapy for acute intoxication cases requires a specialized strategy. Acute poisoning can present with a wide variety of signs and symptoms, which include dysrhythmias, decreased blood pressure, miosis, and disturbance of the central nervous system (CNS). Moreover, in some clinical cases, poisoning can be high enough to cause the failure of multiple organ systems [19]. It was stated that health outcomes would be favorable in the majority of instances if the symptoms of the poisoning were recognized and identified early on, and if proper diagnostic and supportive treatment were commenced as quickly as possible [20]. The surveillance system in Saudi Arabia is working on an ongoing systematic collection of health data that are essential to the planning, implementation, and evaluation of public health practice. We believe that statistical tests are very important to evaluate the collected data and provide a possible explanation by either accepting or rejecting the null hypothesis. The analyzed surveillance data are crucially important for different purposes, such as recognizing epidemiological trends and finding the factors associated with the increase or decrease of incidents and/or deaths. Moreover, the analyzed surveillance data might help to identify high-risk groups as well as the geographical regions requiring control measures. Therefore, there is a great need for clarifying the profile of acute chemical poisoning that might assist authorities in the development of a strategy for its prevention and with control calls to be conducted in every nation and area through extensive epidemiologic research [21]. At the moment, several research projects on chemical intoxication in a variety of Saudi Arabian populations are being carried out. However, to the best of our knowledge, there has not been any recent research conducted in detail on the topic of acute chemical poisoning among children in different regions of Saudi Arabia. In this study, we examined the epidemiological trends of acute chemical poisoning among children of different ages in five different regions of Saudi Arabia. In our study, we included the most important aspects of acute chemical intoxication in detail. For instance, patients’ characteristics (gender, age, and nationality), the type of chemicals to which children had been exposed to, places of incidence (home and other), types of exposure (intentional and unintentional), signs and symptoms, and patients’ outcomes postacute chemical poisoning occurrence.

## 2. Materials and Methods

This research was a 3-year retrospective evaluation of acute chemical intoxication incidents that occurred from 2019 to 2021 in Saudi Arabia. As part of a system of epidemiological surveillance established in 2000 for chemical poisoning cases, the incidents from the hospitals are reported to the general directorate of environmental health department, Ministry of Health (MOH), Riyadh, Saudi Arabia (Source of data in this study). Microsoft Excel was used to save all cases. The director of the general directorate of environmental health created a data-collecting sheet to collect individual and sociodemographic data, such as age (including all age groups, either children or adults), gender, nationality, exposure route (oral, inhaled, intravenous, etc.), symptoms of intoxication, the city where the poisoning occurred, poison category (detergents, disinfectants, pesticides, fuels, etc.), chemical name, cause of poisoning (intentional or accidental), whether specific antidotal therapy for poisoning was administered, and patient outcome (whether recovered or died). Data from a three-year (2019 to 2021) period were collected from the general directorate of environmental health, MOH, Riyadh, Saudi Arabia. We were unable to use the data before 2019 as they were used by others for other purposes. The source of the data mentioned above provided all poisoning cases from different cities in Saudi Arabia. According to the geographical locations of the cities, and to minimize confusion, we classified them into five regions: northern, southern, eastern, western, and central. This study was approved by the Institutional Review Board (IRB), general administration for research and studies, MOH, Riyadh, Saudi Arabia, with reference number: 22-33E. Statistical package for the social sciences (SPSS) software was used for the statistical analysis of the data (International Business Machines (IBM) SPSS Statistics 21, IBM Corporation, Armonk, NY, USA, 2014). Simple frequency tables were used to construct descriptive statistics. One-way analysis of variance (ANOVA) was used to analyze three or more independent data groups. The posthoc Tukey’s multiple comparison test was used to determine which data groups were significantly different if the one-way ANOVA test showed significance. Regression tests were performed using factor characteristics (gender, age, nationality, place of incidence, and circumstances of exposure) as independent variables and the number of acute chemical poisoning cases recorded as the dependent variables. Therefore, regression tests were used to analyze the relationship between the independent variables and the dependent variables between the two groups. The level of statistical significance was set at *p <* 0.05.

## 3. Results

For 2019, the number of acute chemical intoxication cases in the regions was as follows: northern region: 365 (38.2%), southern region: 156 (16.3%), eastern region: 111 (11.6%), western region: 187 (19.6%), and central region: 135 (14.1%) (Figure 1A). In 2020, acute chemical intoxication cases in the northern region were also the highest, with 429 (40.3%), compared to the southern region: 117 (11%), eastern region: 81 (7.6%), western region: 143 (13.4%), and central region: 292 (27.4%) (Figure 1A). Similar to the two previous years, acute chemical intoxication cases in the northern region recorded the highest rate in 2021 compared to the other regions; in the northern region, there were 416 (41.8%) cases compared to 126 (12.6%) in the southern region, 72 (7.2%) in the eastern region, 164 (16.5%) in the western region, and 215 (21.6%) in the central region (Figure 1A). Altogether, over the three years, respectively, from 2019 to 2021, the mean of the total chemical poisoning frequency in the northern region (403) was significantly higher compared to the mean of the southern region (133), (*p* ≤ 0.001), the eastern region (88), (*p* ≤ 0.001), the western region (164), (*p* ≤ 0.001), and the central region (214) (*p* ≤ 0.002) (Figure 1B). According to the general authority for statistics in Saudi Arabia, the population of the northern region is (320,524), the southern region is (4,980,577), the eastern region is (7,995,750), the western region is (9,567,766), and the central region is (9,686,219) [22]. Therefore, the incidence rate of acute chemical intoxication was high in the northern region (1.25/1000 people/year) compared to the four other regions, respectively (the southern region; 0.02/1000 people/year, the eastern region; 0.01/1000 people/year, the western region; 0.01/1000 people/year, and the central region; 0.02/1000 people/year).

The features of the occurrences of acute chemical poisoning that were recorded between 2019 and 2021 are shown in (Table 1). In 2019, there was a total of 954 incidents, while in 2020, there were 1062 cases documented; in 2021, there were 993 cases. In comparison to the cases reported in the other age groups, the frequency of chemical poisoning among children aged 1 to 5 years was the highest from 2019 to 2021, accordingly (752, 863, and 686). Moreover, the Saudi population is 32,550,836 compared to the non-Saudi population of 10,922,646, based on the latest statistics [22]. Therefore, the incidence rate of chemical intoxication over the three-year period among Saudis is 0.04 per 1000 people/year, while in non-Saudis, it is 0.007 per 1000 people/year. On the other hand, the number of incidents involving chemicals that occurred in a house was much higher than the number of accidents that took place in other places. In addition, the circumstances of acute chemical intoxication showed that unintentional chemical events were the most prevalent type of chemical exposure throughout the last three years compared to acute intentional events (Table 1).

From 2019 to 2021, the type of acute chemical poisoning was significantly associated with gender (R = 0.99, *p* = 0.0003) (Table 2). When compared to females, the frequency of acute poisoning by all types of chemicals, such as organic solvents, disinfection agents, rodenticide, insecticide, cleaning agents, and unknown substances, was greater among men (*n* = 1701, 56.5%) than it was among females (*n* = 1308, 43.4%) (Table 2). It was interesting to see a disparity in the frequency of acute chemical intoxication across all age groups. The frequency of acute chemical poisoning cases was found to be significantly higher in children aged 1–5 than in those less than a year old (R = 0.88, *p* = 0.016) (Table 2). In addition, adolescents aged 13–19 demonstrated higher chemical toxicity cases compared to children aged 6–12, but it was not statistically significant (R = 0.69, *p* = 0.07) (Table 2). The majority of cases of chemical poisoning across all age categories were caused by organic solvents in addition to disinfectant agents. On the other hand, acute chemical poisoning occurred at a much higher rate among Saudi nationals than among foreigners, as seen in Table 2; however, this was not statistically significant (R = 0.74, *p* = 0.057). Most types of poisoning were associated with greater rates among Saudis; however, organic solvents were the most concerning (22.2% vs. 6.07%). Interestingly, there was a statistically significant difference (R = 0.79, *p* = 0.043) between exposure to acute chemical intoxication at home and elsewhere (Table 2). Furthermore, the majority of the cases of acute chemical poisoning were the result of unintentional accidents (*n* = 2120, 70.4%) and were significantly higher (R = 0.85, *p* = 0.024) compared to purposeful acts (*n* = 134, 4.4%).

It was important to evaluate the signs and symptoms of postacute chemical intoxication exposure to see whether this might cause major side effects. The most often reported symptom following acute chemical poisoning exposure in the previous three years was vomiting (20.9%). The next most common symptoms were nausea (6.5%), abdominal pain (4.3%), and breathing difficulties (3%). Other symptoms, such as fever, lack of appetite, headache, dizziness, muscular soreness, skin rash, and constriction, were also observed, albeit at a far lower incidence (Figure 2A).

Treatment intervention was documented in the majority of patients in our investigation, as indicated in Figure 2B. Antidote pharmacological treatment was administered to 1323 patients (43.9%), including activated charcoal (*n* = 664, 50.1%), N-Acetylcysteine (*n* = 383, 28.9%), antihistamine (*n* = 127, 9.5%), naloxone (*n* = 39, 2.9%), atropine (*n* = 45, 3.4%), and fomepizole (*n =* 65, 4.9%) (Figure 2B). On the other hand, the frequency of patients who received supportive treatment after chemical poisoning was 225 (7.4%), and there were 900 patients (29.9%) who did not need treatment (Figure 2B).

In terms of patient outcomes, there were 1554 (51.6%) who were discharged from the emergency department following treatment, and 618 (20.5%) children who were hospitalized in the pediatric ward (Figure 2C). Moreover, there were 268 (8.90%) admitted to the pediatric intensive care unit (PICU). In the PICU, eight deaths were recorded (0.2%) (Figure 2C).

## 4. Discussion

According to the findings of our study, there has been a steady prevalence of acute chemical intoxication among children over the last three years (2019 to 2021) in Saudi Arabia, which might mean that there is an awareness of chemical poisoning among the population. Despite the fact that there was not a decrease in the prevalence of acute chemical intoxication over the three years, the population as a whole is experiencing continual changes in both their way of life and their social conduct, which may be the source of this steady prevalence [23]. In addition, it might be because of the lockdown that was implemented during the COVID-19 pandemic. Another possible explanation is that the country of Saudi Arabia is experiencing rapid population growth.

According to the findings of our research, the incidence rate of acute chemical poisoning has been much higher in the northern part of Saudi Arabia in comparison to the other regions during the last three years, correspondingly. This might likely be due to a lack of awareness in the population living in the northern region, which, in turn, makes the incidence of hazardous exposure to those chemicals even more common, although harmful exposure is unavoidable. Since Arabic is the national language of Saudi Arabia, there is another possibility that another contributing factor to this rise is the practice (adopted by some businesses) of labeling chemical products in the English language. As a result, some Saudi parents are unable to comprehend the directions for using the products properly.

In contrast to a tertiary care hospital in India [24] and an epidemiological study of acute chemical intoxications in the Jeddah region between 2011 and 2015 [25], our study revealed that males were highly represented in the number of acute chemical intoxication cases compared to females from 2019 to 2021, and this is statistically significant. Furthermore, in contrast to our data, other studies, including the National Guard Hospital in Jeddah City, Saudi Arabia [26], as well as those in the cities of Al-Qassim and Riyadh, Saudi Arabia [27,28], India [29], and Malaysia [30] have all reported a higher rate among women. This pattern may have to do with who the chemicals are being marketed to, where these cases have been reported, how wealthy the people involved are, how hierarchical the society is, or how violent it is.

Additionally, the findings of our study revealed that the age group of children between one and five years old was the most poisoned by chemicals. This is in line with the results of earlier Saudi studies carried out in Riyadh City [27], Jeddah’s National Guard Hospital [26], and Al-Qassim City [28], which all presented similar findings. This might be due to the adventurous tendencies, hyperactivity, and natural curiosity of children, who have the propensity to put everything they come into contact with into their mouths and are unable to differentiate between harmful and safe things [16], or it could be because there is not enough supervision in place. Additionally, the majority of chemical poison containers do not have child-resistant caps [26,27,28]. These results bring up serious concerns about the need to counsel parents on how to avoid incidents of chemical poisoning among children. However, recent research conducted in India found that most incidences of chemical poisoning occurred in people between the ages of 21 and 30 [31]. A possible explanation for this discrepancy is that over 70% of the Indian cases were caused by suicide, which is more common in adults than in children [31]. Other studies in India [29] and Malaysia [30] found a similar pattern, with adults reporting being poisoned at a greater rate than children.

Based on our findings, there was a relationship between age and the specific chemical that was consumed. Organic solvents and disinfectants posed a greater risk of poisoning individuals of all age groups than any other chemical substance. This might be attributable to societal income differences and/or contaminated hygiene by either organic solvents or disinfectant agents. Our research is in line with a previous study, which reported that organic solvents are a major contributor to child poisoning in developing nations [32,33]. Furthermore, the detergents used for disinfecting were shown to be the cause of 21% of poisoning cases in children less than 12 years old in research performed in Bangladesh, Colombia, Egypt, and Pakistan [34]. On the other hand, pesticides were shown to be the most frequent poisons in other studies [16,29,35]. Economic factors possibly account for these discrepancies. Moreover, pesticides may be one of the most prevalent sources of chemical poisoning in agriculturally based communities.

We also observed that the majority of instances of acute chemical intoxication were among Saudis, which most likely represents the demographic distribution pattern in Saudi Arabia. This conclusion accords with the findings of previous studies that have been carried out in Saudi Arabia [26,27,28]. The incidence rate of acute chemical intoxication is higher in Saudis (0.04 per 1000 people/year) compared to non-Saudis (0.007 per 1000 people/year) from 2019–2021) in Saudi Arabia; however, this might be because the Saudi population is higher than the non-Saudis population. Furthermore, the exact number of Saudi and non-Saudi children is not known, which might affect this outcome. Further studies are needed to evaluate the incidence rate of chemical poisoning among an exact number of children in Saudi Arabia with a larger sample size of the population.

Additionally, our research revealed that there is a significant increase in the number of instances of acute chemical poisoning that are reported from home compared to those reported from other places. This is in line with previous studies that found the same outcomes [36,37]. This might be due to the prevalence of household chemical products, the ease with which children can access a storage cabinet, and the attractiveness of certain product packaging, all of which can contribute to the poisoning of children [36].

Based on the data we gathered, we determined that the proportion of cases of acute accidental chemical poisoning was significantly higher than that of acute intentional poisoning. This is consistent with another study that reported male (56.3%) and female (43.7%) children were more likely to have acute unintentional chemical poisoning [38]. This may be because of a rise in the potential causes of accidents, such as chemical accessibility, improper chemical storage, insufficient adult supervision, and daytime parental work [38]. On the other hand, our acute intentional chemical poisoning reported cases (134 (4.4%)) were lower than the other cases that have been reported by other studies [26,27,29]. Therefore, differences in research demographics, reporting locations, or reporting cases might all play a role in these inconsistencies.

The analysis of the medical data from our study revealed that vomiting was the most common symptom of acute chemical poisoning over the three-year period. After that, the frequency of symptoms such as nausea, abdominal pain, and breathing difficulty after acute chemical poisoning was greater compared to the frequency of other symptoms. The findings of our study agreed with those from a study that gathered 647 telephone inquiries on liquid detergent capsule toxicity and found that the most frequent symptoms included vomiting (24.1%), coughing (4.1%), nausea (3.5%), tiredness (1.7%), and skin rash (1.7%) [39].

The management of chemical poisoning incidents differed from one case to another based on the patient’s health, the kind of poisoning, and the period of exposure [40]. According to the findings of this study, the majority of acute poisoning cases in which an antidote was administered were handled in the hospital by decontamination with activated charcoal (50.1%) or N-acetylcysteine (28.9%). According to earlier studies, activated charcoal by itself is a good therapy for poisoning cases that arrive at a hospital within an hour of the incident as it can reduce the absorption of the chemicals in the stomach and intestine [41]. Furthermore, 29.9% of acute chemical intoxication cases admitted to the hospital did not need treatment. This might be due to the accidental ingestion of small amounts or minimally toxic substances or the awareness of parents of applying the first aid procedures immediately after chemical poisoning.

With regards to the ultimate patient outcomes after exposure to acute chemical intoxication in this study, it was found that the vast majority of patients (51.6%) were treated in the emergency departments and sent home after a period of surveillance to ensure their health was stable. The clinical severity of the majority of cases in our study was modest, with (20.5%) admitted to the pediatric ward and only (8.9%) admitted to the PICU. Regrettably, eight patients (0.2%) died after hospitalization. This is very low compared to some previous studies, which found that the average poisoning mortality rate in developed countries was 1%, and it was 3–5% in developing countries [42]. Moreover, nine fatalities (2.2%) were observed following chemical poisoning in research conducted in Al-Qassim Province, Saudi Arabia [28], while 11 deaths (1.1%) were recorded in another study performed in Jeddah City, Saudi Arabia [25].

In terms of the limitations in our study, it is possible that information bias played a role in these findings due to either the absence or nonreporting of data. According to the source of the data, this concern is divided into two parts. The first refers to the poisoned person. Children under the age of five are the majority of individuals who experience poisoning. As a result, most chemical exposures to children occur away from parents’ view, either in storage areas or in other parts of the house where the material is present. Consequently, the doctor’s decision to admit the child to the hospital depends on the symptoms and the type of chemical to which the child was exposed. This relies on whether the chemical substance was present at the site of exposure or whether the source of the poisoning was known. This is one of the challenges: not knowing the chemical substance, resulting in not reporting the data. The second is the available analysis of chemicals in special poison centers. Most of the incendiary, cleaning, or sterilizing materials can not be analyzed in the laboratories, and this might contribute to not knowing the type of chemical substance, resulting in not reporting the data or missing data on the names of chemical agents. Therefore, to minimize the nonreporting of data in the future, we are suggesting that the surveillance system could be improved by responding quickly and efficiently to incidents, reporting chemical poisoning incidents in detail from emergency departments, and developing laboratory analysis to identify substances relating to poisoning. Furthermore, chemical poisoning prevention work can be performed by activating the awareness and educational role of the community based on the data from cases of poisoning, which might help with determining the most common chemical substances. This can be performed by distributing and publishing information using video clips in the primary health care centers, as well as participating in events for international days with other partner sectors to provide awareness lectures or exhibitions in schools, commercial gatherings, event centers, etc.

Despite the fact that all fundamental services, such as education, health, etc., are supplied at the same level in all the regions of Saudi Arabia, the exact reason behind the highest cases of acute chemical poisoning among children in the northern region is unknown. However, there may be a relative variation in the level of knowledge among the population in different regions and among parents in terms of dealing with the chemicals present inside or outside the house. Furthermore, the size of the population, economic profile, and living conditions are different among the population in Saudi Arabia. All of these aspects might contribute to the acute chemical poisoning differences among children in the regions of Saudi Arabia. Therefore, further investigations are needed to identify why the northern region of Saudi Arabia has the highest cases of acute chemical poisoning among children.

## 5. Conclusions

Our study sheds light on the epidemiological trends of acute chemical poisoning among children over a recent three-year period in Saudi Arabia. In this study, we evaluated the recorded data on acute chemical intoxication with many types of chemical substances regarding children from different regions. Gender, age, nationality, the place where the poisoning occurred, and the circumstances of the poisoning (either accidental or unintentional) were included in this evaluation. We found that the northern region of Saudi Arabia recorded the highest prevalence rate of acute chemical poisoning compared to the other four regions. Our study revealed that males were highly represented in the number of acute chemical intoxication cases compared to females from 2019 to 2021. Moreover, children between one and five years old were the worst hit by chemical substances. On the other hand, organic solvents and disinfectants posed a greater risk of poisoning individuals in all age groups compared to any other chemical substances. Additionally, our research revealed that there was a significant increase in the number of instances of acute chemical poisoning reported from home compared to those reported from other places. Finally, the recorded cases of accidental acute chemical poisoning were significantly higher than intentional acute poisoning events. Awareness is highly needed among parents and household cleaners, as well as the proper storage of chemicals. A child’s knowledge of poisoning may be enhanced by teaching them about the dangers at an early age. Additionally, physicians must provide more thorough notes in the electronic medical records (EMR) system in the hospitals. Therefore, the outcomes of our study may drive better future service designs and health policies, as well as health strategies. On the other hand, dangerous chemical products should be monitored in the marketplace by the regulatory sectors that license these compounds. Furthermore, providing these regulatory sectors with the analyzed data from studies may lead to the development of collaborative working partnerships with the Ministry of Health to collaborate and develop ways to reduce the occurrence of chemical poisoning among all individuals in society.

## Figures and Tables

**Figure 1 children-10-00295-f001:**
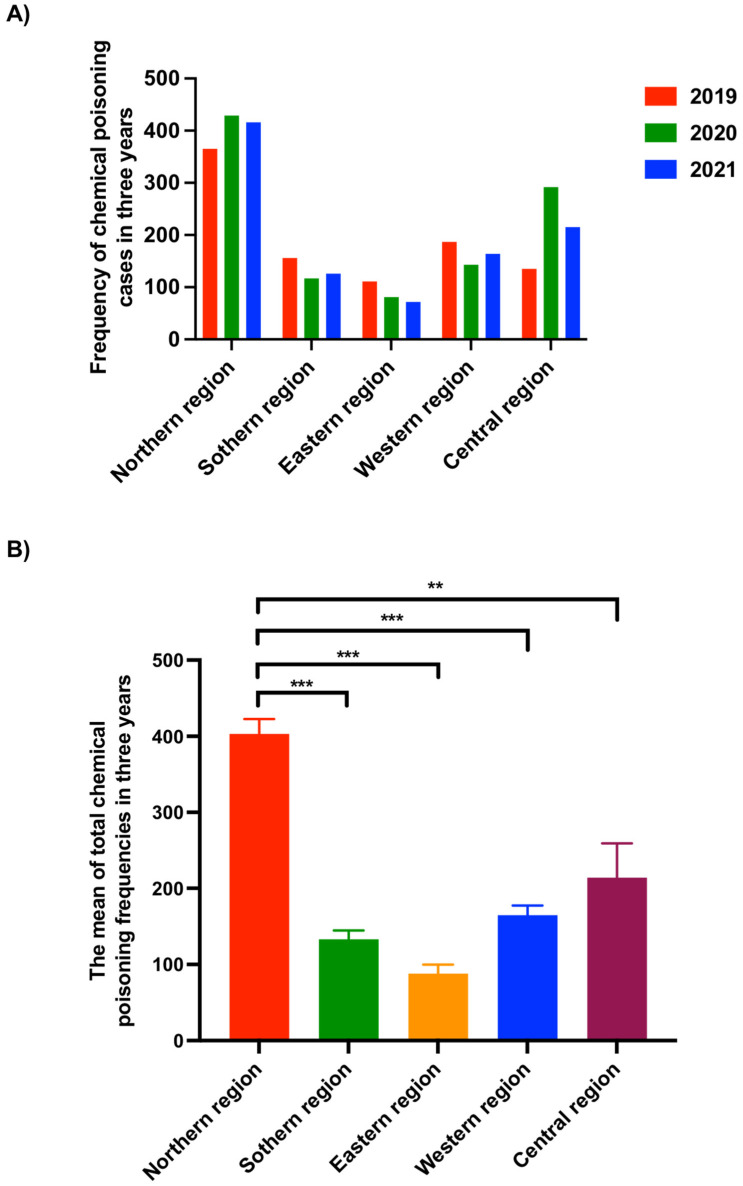
The frequency of acute chemical intoxication cases from 2019 to 2021 in the five regions of Saudi Arabia. Statistical analysis was performed using a one-way ANOVA followed by a posthoc Tukey’s multiple comparison test to identify which data groups were significantly different. ** *p* < 0.01, *** *p* < 0.001. (**A**) The frequency of acute chemical poisoning cases in three years period individually. (**B**) The mean of total acute chemical poisoning frequencies from 2019 to 2021.

**Figure 2 children-10-00295-f002:**
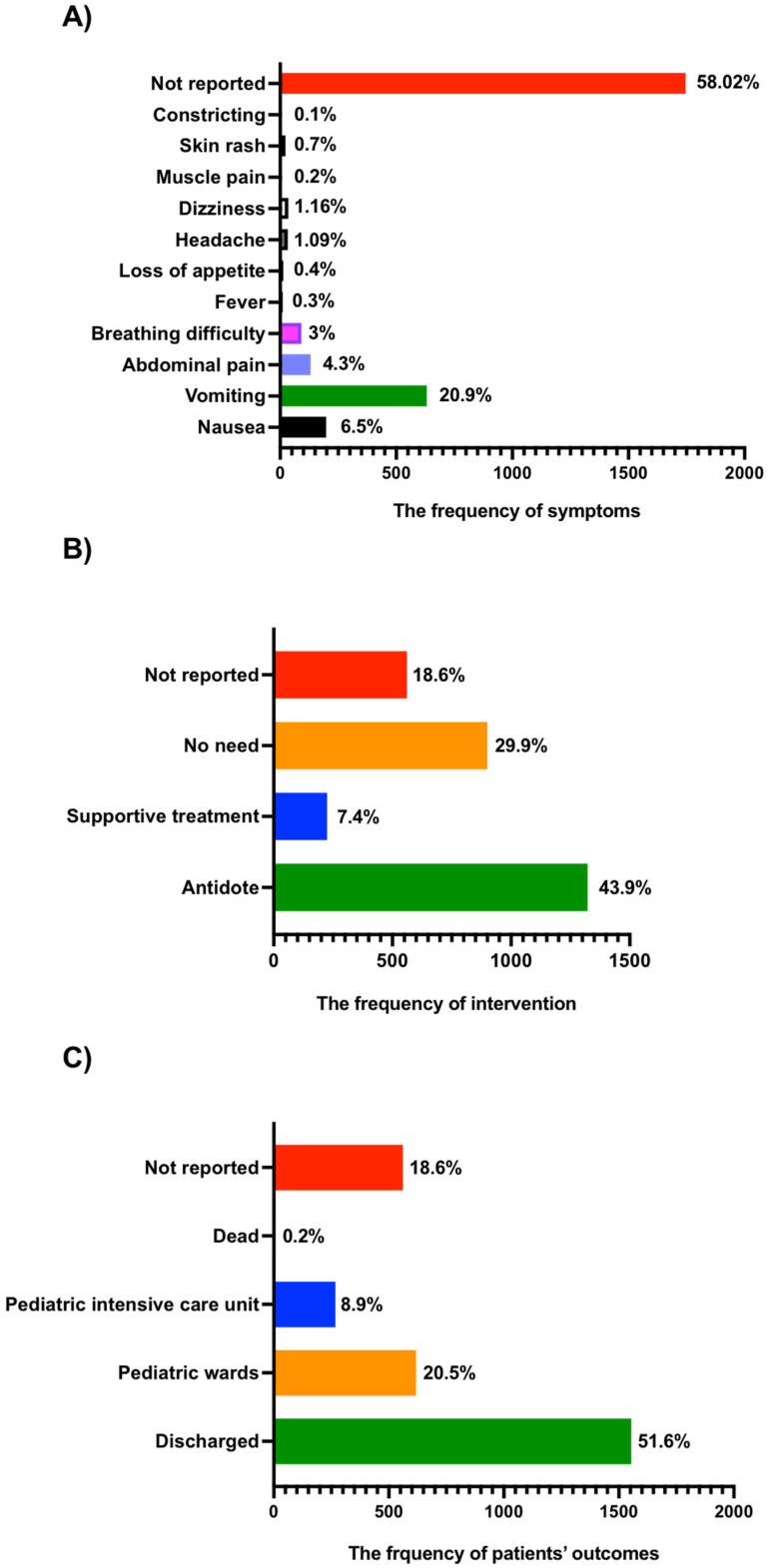
Signs and symptoms, medical interventions, and the outcomes of patients postacute chemical poisoning reported cases from 2019 to 2021 within this study. The frequency of signs and symptoms (**A**), medical interventions (**B**), the outcomes of patients (**C**) postacute chemical poisoning reported cases from 2019 to 2021 within this study.

**Table 1 children-10-00295-t001:** Characteristics of chemical intoxication cases within this study.

Year of Incidence	2019	2020	2021	Total
Variables	N = 954 (%)	N = 1062 (%)	N = 993 (%)	N = 3009 (%)
(Gender)				
Male	550 (57.6)	591 (55.6)	560 (56.3)	1701 (56.5)
Female	404 (42.3)	471 (44.3)	433 (43.6)	1308 (43.46)
(Age)				
<1 year	49 (5.1)	48 (4.5)	140 (14)	237 (7.8)
1–5 years	752 (78.8)	863 (81.2)	686 (69)	2301 (76.4)
6–12 years	72 (7.5)	59 (5.5)	83 (8.3)	214 (7.1)
13–19 years	81 (8.4)	92 (8.6)	84 (8.4)	257 (8.5)
(Nationality)				
Saudi	864 (90.5)	976 (91.9)	922 (92.8)	2762 (91.7)
Non-Saudi	90 (9.4)	86 (8)	71 (7.1)	247 (8.2)
(Place of incidence)				
Home	845 (88.5)	956 (90)	901 (90.7)	2702 (89.7)
Others	109 (11.4)	106 (9.9)	92 (9.2)	307 (10.2)
(Circumstances of exposure)				
Intentional	35 (3.6)	60 (5.6)	39 (3.8)	134 (4.4)
Unintentional	530 (55.5)	751 (70.7)	839 (84.4)	2120 (70.4)
Unknown	389 (40.7)	251 (23.6)	115 (11.5)	755 (25.0)

**Table 2 children-10-00295-t002:** The cases of chemical poisoning from 2019 to 2021 are classified by gender, age group, and other variables. Regression analysis was performed to identify if the relationship between the independent variables and the dependent variable between the two groups was significantly different. R square = regression value.

Variables	Organic Solvents	Disinfectant	Rodenticide and Insecticide	Cleansing Substances	Unknown	Total	R Square (*p-*Value)
	N (%)	N (%)	N (%)	N (%)	N (%)		
(Gender)							
Male	370 (21.7)	385 (22.6)	108 (6.3)	114 (6.7)	724 (42.5)	1701	0.99
Female	259 (19.8)	299 (22.8)	94 (7.1)	70 (5.3)	586 (44.8)	1308	(0.0003)
(Age)							
<1 year	29 (12.2)	55 (23.2)	18 (7.5)	24 (10.1)	111 (46.8)	237	0.88
1–5 years	498 (21.6)	505 (21.9)	143 (6.2)	140 (6.08)	1015 (44.1)	2301	(0.016)
6–12 years	58 (27.1)	45 (21.02)	20 (9.3)	12 (5.6)	79 (36.9)	214	0.69
13–19 years	30 (11.6)	79 (30.7)	18 (7.0)	8 (3.1)	122 (47.4)	257	(0.07)
(Nationality)							
Saudi	615 (22.2)	639 (23.1)	176 (6.3)	165 (5.9)	1167 (42.2)	2762	0.74
Non-Saudi	15 (6.07)	45 (18.2)	22 (8.9)	19 (7.6)	146 (59.1)	247	(0.057)
(Place of incidence)							
Home	526(19.4)	661 (24.4)	195 (7.2)	170 (6.2)	1150 (42.5)	2702	0.79
Others	85 (27.6)	23 (7.4)	4 (1.30)	13 (4.2)	182 (59.2)	307	(0.043)
(Circumstances of exposure)							
Intentional	9 (6.7)	29 (21.6)	9 (6.7)	4 (2.9)	83 (61.9)	134	0.85
Unintentional	425 (20.04)	524 (24.7)	136 (6.4)	153 (7.2)	882 (41.6)	2120	(0.024)
Unknown	181 (23.9)	130 (17.2)	52 (6.8)	28 (3.7)	364 (48.2)	755	

## Data Availability

The data presented in this study are available on request from the corresponding author.

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
