# Peer review of "Epidemiological Trends of Acute Chemical Poisoning among Children over a Recent Three-Year Period in Saudi Arabia"

_children, 2023, doi:10.3390/children10020295_

Round 1

Reviewer 1 Report

Manuscript ID: children-2017475

Epidemiological trends of chemical poisoning among children over a recent three-year in Saudi Arabia

The authors make use of the surveillance system for poisonings in Saudi Arabia and aim to analyze data with respect to reported intoxications of children. While the data base might be a valuable tool for monitoring of poisonings and the study of associations this paper unfortunately does not make most of it. I follow some examples which point to limitations of the paper:

Introduction.

This chapter does not introduce the research question in a straight forward way. Very general statements are based on very specific references e.g. [1,2] and statement seem not to be correctly reported: e.g

·         “Acute poisoning referral rates to emergency rooms vary widely among countries, ranging from 0.076% to 0.7% annually [4,5].” Besides that the references [4,5] are quite outdated being more than 20 years old (!) it is not clear which rate is meant. % of population, % of admission to hospitals? % of poisoning?

·         Share of suicides by pesticides: [7] Gunnell et al. give 30%. The overall range is larger.

·         WHO report [3] gives a rate of 2 per 100.000 children – again 20 years ago! - which is somehow transferred to 11.6 % by the authors. Other sources for mortality rates are obviously wrongly cited, e.g. Sahin et al ref [11] states „None of our patients died as a result of poisoning.”

Materials and methods

This chapter is insufficient. We do not learn anything about the epidemiological surveillance and the reporting system. Who is reporting? What is the coverage? Is there a follow up for cases? Does the data include calls to poisoning control center or hospitals? Why only 3 years considered? What data was included? Which agents are reported/grouped? Which age groups were selected? Why were regression technique not applied to study multiple influences and to control for confounding effects? As it was aimed to compare regions what regions were chosen. How do they differ?

Results

The wording and the data shown mismatch. E.g.

·         “The incidence rate of chemical intoxication was high in northern regions compared to the other four regions” but no incidence rates are shown. Instead the figures show the distribution of cases across the regions. As we have not learned about the size of the population of the regions there is no way to calculate incidence rates.

·         “Chemical poisoning occurred at a much greater rate among Saudi nationals than among foreigners, as seen in table” Again this rate is not given but the percentage of Saudi nationals in all cases. Given, that most people in Saudi Arabia are Saudi nationals this should not be a surprise.

Discussion and conclusion

This chapters lacks mentioning any shortcomings. No critical appraisal of the data source e.g. missing data on agents for more than 40% of cases. Effects of not controlling for confounding factors are not considered nor of region’s differences e.g. with respect to size of population, economic profile, living conditions.

Author Response

Dear Editor,

We are submitting our revised manuscript entitled “Epidemiological trends of chemical poisoning among children over a recent three-year in Saudi Arabia” to Children for evaluation for publication. Reference number for this manuscript is children-2017475.

Thank you for giving me the opportunity to submit a revised draft of my manuscript to “Children journal”.

I appreciate the time and effort that you and the reviewers have dedicated to providing your valuable feedback on my manuscript. I am grateful to the reviewers for their insightful comments on my paper. I have been able to incorporate changes to reflect most of the suggestions provided by the reviewers.

Here is a point-by-point response to the reviewers’ comments and concerns. 

Sincerely,

Mohammed Merae Alshahrani, PhD.

Assistant Professor

Najran University, Saudi Arabia  

Point-by-Point Response to Comments: (children-2017475):

Reviewer 1

Introduction.

        Acute poisoning referral rates to emergency rooms vary widely among countries, ranging from 0.076% to 0.7% annually [4,5].” Besides that, the references [4,5] are quite outdated being more than 20 years old (!) it is not clear which rate is meant. % of population, % of admission to hospitals? % of poisoning?

         Share of suicides by pesticides: [7] Gunnell et al. give 30%. The overall range is larger.

          WHO report [3] gives a rate of 2 per 100.000 children – again 20 years ago! - which is somehow transferred to 11.6 % by the authors? Other sources for mortality rates are obviously wrongly cited, e.g., Sahin et al ref [11] states „None of our patients died as a result of poisoning.”

      The authors thank the learned reviewer for these comments. All of these comments been updated in the introduction section of revised manuscript.

Materials and methods

This chapter is insufficient. We do not learn anything about the epidemiological surveillance and the reporting system. Who is reporting? What is the coverage? Is there a follow up for cases? Does the data include calls to poisoning control centre or hospitals? Why only 3 years considered? What data was included? Which agents are reported/grouped? Which age groups were selected? Why were regression technique not applied to study multiple influences and to control for confounding effects? As it was aimed to compare regions what regions were chosen. How do they differ?

       The authors thank the learned reviewer for these important comments. This research was a 3-year retrospective evaluation of chemical intoxication incidents that occurred from 2019 to 2021 in Saudi Arabia. As part of a system of epidemiological surveillance established in 2000 for chemical poisoning cases, the incidents from the hospitals are reported to the general directorate of environmental health department, Ministry of Health (MOH), Riyadh, Saudi Arabia (Source of data in this study). Microsoft Excel is used to save all cases. The system's goal is to keep track of the short- and long-term damage that chemical manufacture, storage, transport, and usage causes to people and the environment. The director created a data collecting sheet to collect individual and sociodemographic data, such as age (including all age groups either children or adults), gender, nationality, exposure route (oral, inhaled, intravenous, etc.;), symptoms of intoxication, the city where the poisoning occurred, poison category (detergents, disinfectants, pesticides, fuels, etc.), chemical name, precipitating cause of poisoning (intentional or accidental), whether specific antidotal therapy for poisoning was administered, and patient outcome (whether recovered or died). Data of recent three years (2019 to 2021) were collected from the general directorate of environmental health, MOH, Riyadh, Saudi Arabia. We were unable to use the data before 2019 as they have been used by others for other purposes. The source of data, which mentioned above, provided all poisoning cases from different cities in Saudi Arabia. According to cities' geographical locations and to minimize the confusion, we classified them into five regions: northern, southern, eastern, western and central. Furthermore, as a learned reviewer suggested, regression test has been applied to the data. Now, all of these comments been incorporated the materials and methods section of the revised manuscript.

Results

The wording and the data shown mismatch. E.g.

     The incidence rate of chemical intoxication was high in northern regions compared to the other four regions”, but no incidence rates are shown. Instead, the figures show the distribution of cases across the regions. As we have not learned about the size of the population of the regions there is no way to calculate incidence rates.

      The authors thank the learned reviewer for these important comments. According to general authority for statistics in Saudi Arabia, the population of the northern region is (320,524), the southern region is (4,980,577), the eastern region is (7,995,750), the western region is (9,567,766), and the central region is (9,686,219) (Reference 24 in the revised manuscript).Therefore, the incidence rate of chemical intoxication in northern region was (1.25/1000 people-year) compared to the other four regions respectively, in which (southern region; 0.02/1000 people-year, eastern region; 0.01/1000 people-year, western region; 0.01/1000 people-year and central region; 0.02/1000 people-year). Now, all of these comments been incorporated the result section of the revised manuscript.

     Chemical poisoning occurred at a much greater rate among Saudi nationals than among foreigners, as seen in table” Again this rate is not given but the percentage of Saudi nationals in all cases. Given, that most people in Saudi Arabia are Saudi nationals this should not be a surprise.

      The authors thank the learned reviewer for these important comments. Yes, we do totally agree with the reviewer about this point. The Saudi population is (32,550,836) compared to non-Saudi population (10,922,646) based on the latest statistics (Reference 24 in the revised manuscript). Therefore, the incidence rate of chemical intoxication over a recent three years in Saudis is 0.04 per 1000 people-year while in non-Saudis is 0.007 per 1000 people-year. Despite the incidence rate of chemical intoxication was higher in Saudis (0.04 per 1000 people -year) compared to non-Saudis (0.007 per 1000 people-year) in the recent three years (2019 to 2021) in Saudi Arabia, however, this might because the Saudis population is more than non-Saudis. Furthermore, the exact number of Saudi and non-Saudi children are not known, which might affect this outcome. Further studies are needed to evaluate the incidence rate of chemical poisoning among exact number of children in Saudi Arabia with a larger sample size of population. All of these comments have been mentioned in the result and discussion sections of the revised manuscript.

Discussion and conclusion

This chapters lacks mentioning any shortcomings. No critical appraisal of the data source e.g., missing data on agents for more than 40% of cases. Effects of not controlling for confounding factors are not considered nor of region’s differences e.g., with respect to size of population, economic profile, living conditions.

       The authors thank the learned reviewer for these important comments. In terms of limitations in our study, it is possible that information bias had a role in these findings due to either an absence or not reporting data. According to the source of data, this concern is divided into two parts. The first, referring to the poisoned person. Children under the age of five are the majority of individuals who catch poisonings. As a result, most chemical exposures to children occur away from the parents' view, either in storage areas or in other parts of the house where the material is present. Consequently, the doctor's decision to admit the child to the hospital depends According to the symptoms, the question of what kind of chemical the child was exposed to arises frequently. This relies on whether the chemical substance was present at the site of exposure or whether the source was known. This is one of the challenges of not knowing the chemical substance resulting in not reporting data. The second, available analysis of chemicals in special poison centers. Most of the incendiary, cleaning or sterilizing materials, according to the classification of liquid chemicals, do not have an analysis in the laboratory, and this might attribute to not knowing the type of chemical substance, in which the child was exposed to resulting in not reporting data or missing data on chemical agents' name. Despite the fact that all fundamental services such as education, health, etc., are supplied at the same level in all regions of Saudi Arabia, there may be a relative variation in the level of knowledge among population in different regions and among parents in terms of dealing with chemicals present inside or outside the house. Furthermore, size of population, economic profile, living conditions are different among population in Saudi Arabia. Therefore, all of these aspects might contribute to the chemical poisoning differences among children in Saudi Arabia regions. All of these comments have been mentioned in the discussion section of the revised manuscript.

Reviewer 2 Report

The manuscript is interesting and provides essential information in the respective field. I have some concerns about the content of the Manuscript:

1 In the introduction section, the aim of the study should be clarified better. National and regional data about chemical poisoning among the pediatric population should be introduced.

2 The methodology should be better defined in the materials and methods section. The terminology and the items discussed in the results section should be elaborated effectively in the introduction and material and methods section to make it more reader-friendly.

3 Tables 2 and 3 need to be improved. The content and the graphics could be more explicit.

4 The discussion should be expanded and needs clarification; it is messy and unclear. The sociodemographic issues of chemical poisoning in pediatrics should be discussed. No data regarding the management and the outcome of chemical poisoning are discussed. The authors should implement the section focusing on these topics. Also, they should expand the discussion treating the circumstances and the manner of poisoning with all the related public health and social issues.

Author Response

Dear Editor,

We are submitting our revised manuscript entitled “Epidemiological trends of chemical poisoning among children over a recent three-year in Saudi Arabia” to Children for evaluation for publication. Reference number for this manuscript is children-2017475.

Thank you for giving me the opportunity to submit a revised draft of my manuscript to “Children journal”.

I appreciate the time and effort that you and the reviewers have dedicated to providing your valuable feedback on my manuscript. I am grateful to the reviewers for their insightful comments on my paper. I have been able to incorporate changes to reflect most of the suggestions provided by the reviewers.

Here is a point-by-point response to the reviewers’ comments and concerns. 

Sincerely,

Mohammed Merae Alshahrani, PhD.

Assistant Professor

Najran University, Saudi Arabia  

Point-by-Point Response to Comments: (children-2017475):

Reviewer 2

The manuscript is interesting and provides essential information in the respective field. I have some concerns about the content of the Manuscript:

We are grateful to the learned reviewer for the appreciation and interest of this manuscript.

1- In the introduction section, the aim of the study should be clarified better. National and regional data about chemical poisoning among the paediatric population should be introduced.

The authors thank the learned reviewer for these important points. They have been updated and incorporated in the revised manuscript.

2- The methodology should be better defined in the materials and methods section. The terminology and the items discussed in the results section should be elaborated effectively in the introduction and material and methods section to make it more reader friendly.

The authors thank the learned reviewer for these comments. The materials and methods section has been expanded in the revised version of this manuscript. We have defined all the terminologies throughout the whole revised manuscript.

3- Tables 2 and 3 need to be improved. The content and the graphics could be more explicit.

The authors thank the learned reviewer for these comments. We provided improved (figure 2) instead of the table 3 in the revised version of this manuscript. We are hoping it is more clearer now. For table 2, as it contains a lot of data, so, we provided classified groups of chemical substances, which children were poisoned with. Moreover, gender, age groups, and other variables including poisoning frequencies of Saudis and non-Saudis, the places where intoxication occurred (home or others) and circumstances of chemical poisoning either accidental or intentional events. Unknown chemical substances were mentioned as well as not reported cases in table 2 in the revised manuscript.

4- The discussion should be expanded and needs clarification; it is messy and unclear. The sociodemographic issues of chemical poisoning in paediatrics should be discussed. No data regarding the management and the outcome of chemical poisoning are discussed. The authors should implement the section focusing on these topics. Also, they should expand the discussion treating the circumstances and the manner of poisoning with all the related public health and social issues

The authors thank the learned reviewer for these valuable comments. The discussion section has been expanded and all the comments from the learned reviewer have been incorporated in the discussion section of revised manuscript.

Round 2

Reviewer 1 Report

see attached file

Reviewer 2 Report

I thank the authors for their efforts. They have responded substantively and point by point to my concerns.

Author Response

Thank you for giving me the opportunity to submit a revised draft of my manuscript to “Children journal”.

I appreciate the time and effort that you and the reviewers have dedicated to providing your valuable feedback on my manuscript. I am grateful to the reviewers for their insightful comments on my paper.